# Clonal architecture in mesothelioma is prognostic and shapes the tumour microenvironment

Min Zhang[1,15], Jin-Li Luo[2,15], Qianqian Sun[1,15], James Harber[3,15], Alan G. Dawson [3,4,15], Apostolos Nakas[4], Sara Busacca[3], Annabel J. Sharkey [5], David Waller[6], Michael T. Sheaff[6], Cathy Richards[7], Peter Wells-Jordan[7], Aarti Gaba[3], Charlotte Poile[3], Essa Y. Baitei[3,8], Aleksandra Bzura[3], Joanna Dzialo [3], Maymun Jama[3], John Le Quesne[9], Amrita Bajaj[10], Luke Martinson[3], Jacqui A. Shaw [3], Catrin Pritchard[3], Tamihiro Kamata [3], Nathaniel Kuse[3], Lee Brannan[11], Pan De Philip Zhang[12], Hongji Yang [12], Gareth Griffiths[13], Gareth Wilson[14], Charles Swanton [14], Frank Dudbridge [11], Edward J. Hollox [3] & Dean A. Fennell [3✉]

Malignant Pleural Mesothelioma (MPM) is typically diagnosed 20–50 years after exposure to asbestos and evolves along an unknown evolutionary trajectory. To elucidate this path, we conducted multi-regional exome sequencing of 90 tumour samples from 22 MPMs acquired at surgery. Here we show that exomic intratumour heterogeneity varies widely across the cohort. Phylogenetic tree topology ranges from linear to highly branched, reflecting a steep gradient of genomic instability. Using transfer learning, we detect repeated evolution, resolving 5 clusters that are prognostic, with temporally ordered clonal drivers. *BAP1/*−3p21 and *FBXW7/*-chr4 events are always *early* clonal. In contrast, *NF2/*−22q events, leading to Hippo pathway inactivation are predominantly *late* clonal, positively selected, and when subclonal, exhibit parallel evolution indicating an evolutionary constraint. Very late somatic alteration of *NF2*/22q occurred in one patient 12 years after surgery. Clonal architecture and evolutionary clusters dictate MPM inflammation and immune evasion. These results reveal potentially drugable evolutionary bottlenecking in MPM, and an impact of clonal architecture on shaping the immune landscape, with potential to dictate the clinical response to immune checkpoint inhibition.

[1] Novogene Co., Ltd, Building 301, Beijing, China. [2] Bioinformatics and Biostatistics Support Hub, University of Leicester, Leicester, UK. [3] Department of Genetics and Genome Biology, University of Leicester, Leicester, UK. [4] Department of Cardiothoracic Surgery, Glenfield Hospital, Leicester, UK. [5] University of Sheffield Teaching Hospitals, Leicester, UK. [6] Barts Health NHS Trust, The Royal London Hospital, London, UK. [7] Department of Pathology, Leicester Royal Infirmary, Infirmary Square, Leicester, Leicestershire, UK. [8] Department of Genetics, King Faisal Specialist Hospital and Research Centre, Riyadh, Saudi Arabia. [9] Institute of Cancer Sciences, University of Glasgow, Garscube Estate, Bearsden, UK. [10] Department of Radiology, Glenfield Hospital, Leicester, UK. [11] Department of Health Sciences, University of Leicester, Leicester, UK. [12] Department of Informatics, University of Leicester, Leicester, UK. [13] Southampton Clinical Trials Unit, University of Southampton, Southampton, UK. [14] The Francis Crick Institute, London, UK. [15]These authors contributed equally: Min Zhang, Jin-Li Luo, Qianqian Sun, James Harber, Alan G. Dawson. ✉email: df132@leicester.ac.uk

MPM is a rare, incurable cancer that carries a dismal prognosis. Its incidence is increasing globally despite its primary cause, asbestos exposure, having been known for over 50 years[1]. The global annual incidence is 7–40 per million, with an estimated annual death rate of 38,400[2] which is set to increase due to continued asbestos use. MPM typically grows to enormous volumes averaging 640 ml[3], and exhibits wide variations in tumour aggressiveness. This universally lethal cancer has seen only very limited drug development over the last two decades, with the exception of combination immune checkpoint inhibition[4]. Personalised treatments are lacking for MPM[5], however recent insights into the inter-patient genomic heterogeneity of this cancer have revealed frequent somatic alterations involving the cancer genes *BAP1*, *NF2* and *CDKN2A*[6,7]. Preclinical models have implicated these tumour suppressors as initiators of MPM[8,9] with prognostic significance, particularly when they are concurrent[10].

Major questions remain however regarding the extent and clinical significance of genomic intratumour heterogeneity (ITH) in MPM, and in particular whether constrained, temporal ordering of driver events is required for MPM transformation; critical information that could help to inform the development of precision therapy[5,11].

Clinical phylogenetic studies such as TRACERx lung[12] have shown that clonal neoantigen architecture drives immune surveillance[13] which in turn is a selection pressure leading to immune evasion via immunoediting[14,15]. However, it is not known to what extent ITH modulates host immune surveillance or immune escape in MPM.

In this work, to address these gaps in knowledge, we establish a patient-focused research platform named mesothelioma evolution: deciphering drugable Somatic alterations (MEDUSA), in order to infer a human evolutionary model of MPM tumorigenesis and understand the inter-relationships that exist between genomic ITH, host immune surveillance and clinical phenotype, as well as providing a platform for deriving preclinical models to underpin pharmacogenomic investigations. Here we show that clonal architecture in MPM affects patient survival and composition of the tumour microenvironment.

## Results

We conducted multi-regional whole-exome sequencing of 90 tumour regions from 22 patients (the MEDUSA22 cohort) undergoing routine surgery by extended pleurectomy decortication. Matched DNA isolated from whole blood served as a germline reference for calling somatic events. Sampled regions (4–5 per patient) were anatomically stereotyped across the entire cohort, comprising the apex (R1), pericardium (R2), anterior and posterior costophrenic recesses (R3 and R4) and the oblique fissure (R5; Fig. 1a). Most patients (17/22, 74%) had epithelioid histology with the remaining five patients having biphasic MPM (supplementary data 1 and supplementary fig. 1). The majority of patients were chemonaïve with the exception of one patient, who received neoadjuvant chemotherapy. The cohort exhibited extensive inter-patient and intratumour heterogeneity consistent with a steep gradient of genomic instability (Fig. 1b).

Median sequencing depth across all regions was 261x (range 173–469), median coverage breadth was 99.9% (97–100), with a median tumour purity of 46.5% (18–90; supplementary data 2 and 3). We identified 3523 unique somatic mutations across all the patients in our cohort (single-nucleotide variants and indels) of which 1335 were coding mutations. Total mutation burden was low, 1 mutation/megabase (mt/Mb range 0.4–1.6). Clonal mutation burden was 0.6 mut/Mb (0–0.6) and subclonal mutation burden 0.6 mut/Mb (0.0–1.1). The median number of mutations

per patient in our cohort was 130 mutations (range 86–204) of which a median of 71.5 mutations (42–116) were clonal and 53 mutations (9–120) were subclonal (supplementary fig. 2). The per-patient mutation burden for the cohort is detailed in supplementary data 4.

MPM trees inferred by clonal deconvolution were categorised into two, broad topological classes (supplementary figs. 3A-V and 4, and supplementary data 5). Linear trees were characterised by monophyletic clones, all arising from a common node (or most recent common ancestor). In contrast, branched trees comprised polyphyletic clones arising from subclonal nodes[16,17]. The majority (64%) of MPMs were classed as linear, defined by the absence of subclonal branching (Fig. 1c and supplementary fig. 4), which correlated with both the subclonal mutation (Wilcoxon $p = 0.003$) and SCNA burden (Wilcoxon $p = 0.0034$, Fig. 1d).

We temporally dissected mutational processes operating during early (clonal or ubiquitous somatic events) versus late stages of evolution following subclonal diversification[12,18]. The nucleotide transition C→T was the most common clonal and subclonal base substitution, resulting from deamination of 5-methylcytosine (supplementary fig. 5A). In contrast, subclonal enrichment was most commonly associated with C→A transversion. Mutational signatures inferred using deconstructSigs[19] showed clonal enrichment of clock-like, base excision repair deficiency and single base substitution signatures SBS16, SBS19, SBS32 and SBS39[20], reflecting underlying mutagenesis mechanisms involved in early MPM evolution (supplementary fig. 5A). There was a paucity of subclonal signatures in the cohort with only SBS42 (haloalkanes) showing enrichment during later evolution (supplementary fig. 5B).

Elucidation of cancer genes undergoing positive selection, and therefore operating as likely drivers during early evolution of MPM, was achieved by computing the ratio of clonal non-synonymous to synonymous mutations, normalised by gene size (dN/dS analysis)[21]. Clonal positive selection (dN/dS>1) involved five tumour suppressors *NF2*, *BAP1*, *SETD2* which were independently validated in the mesothelioma TCGA[7], *FBXW7* and *PRELID1* (supplementary fig. 6 and supplementary data 6 and 7). In contrast, subclonal positive selection was only associated with *NF2* (14%).

The majority of driver mutations were predicted as deleterious, loss of function events and were enriched in tumour suppressor genes (supplementary fig. 7A and supplementary data 8). MPMs harboured a median of two clonal driver mutations per patient (range 0–6), with *BAP1* being the most frequent, involving eight patients (36%, comparable to the TCGA's reported 38%). Co-occurrence of multiple driver mutations was common with a median 2 (0–6). Patient MED91 harboured the highest number of drivers, totalling 6.

To establish if clonal driver mutations could be detected (and orthogonally validated) in plasma, we developed patient-specific digital droplet PCR assays to screen the circulating free DNA of 11 patients. Although the rate of detection was low with 4/11 patients positive (36%), overall survival was significantly worse for those patients in whom circulating free DNA could be detected, logrank $p = 0.020$ (Fig. 2a and supplementary data 9–11). The presence of circulating tumour DNA was significantly associated with *NF2* mutation ($p = 0.015$ Fig. 2b), loss of 1p36.11 loss ($p = 0.015$) and/or 6q25.1-2 ($p = 0.018$) and lack of lymph node involvement ($p = 0.024$, Fisher's exact test), supplementary fig. 7B and supplementary data 9–11.

Secondary tumour suppressor hits leading to bi-allelic inactivation of drivers by either mutation and/or copy number loss were more common during early evolution. A median of 1 gene per MPM (0–2) was involved in a clonal double-hit inactivation (supplementary fig. 7C and supplementary data 9). These double-

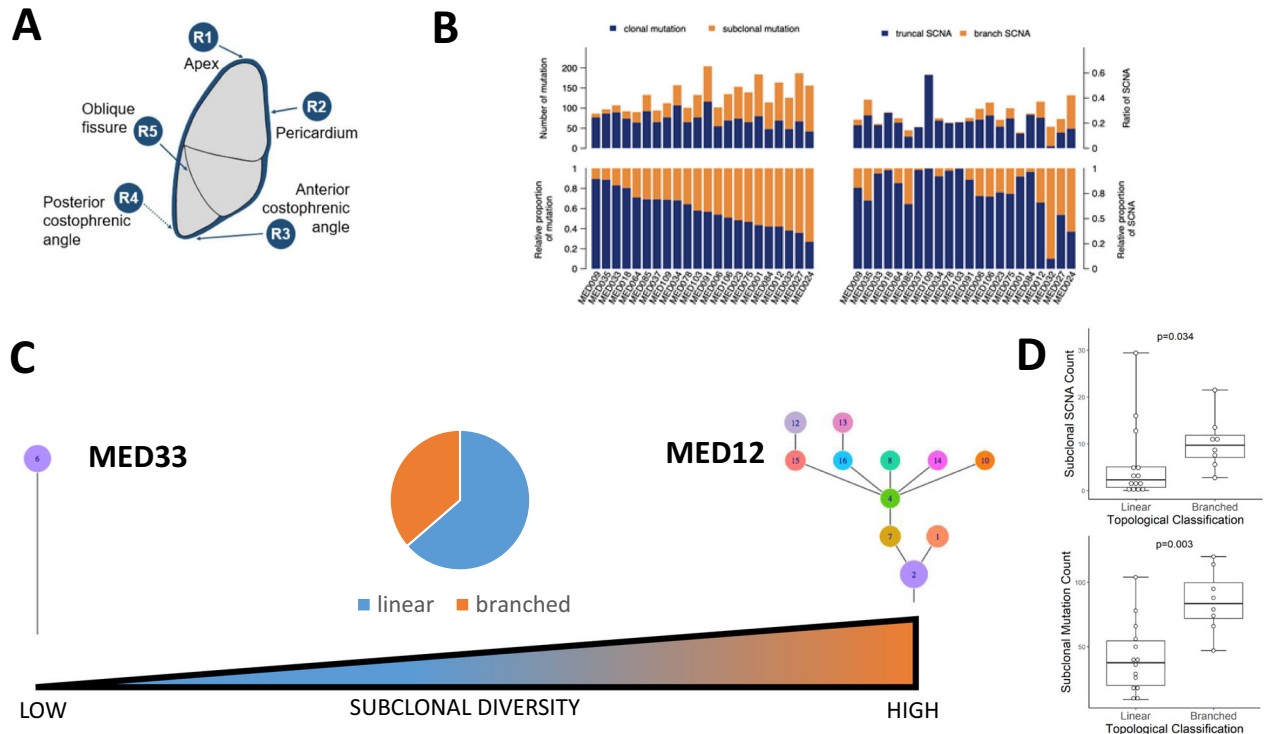

**Fig. 1 Genomic intratumour heterogeneity in MPM. A** Pleural mesothelioma tissue sampling locations were consistent between patients involving the apex (region1, R1), the pericardium (R2), anterior costophrenic angle (R3), posterior costophrenic angle (R4) and the oblique fissure (R5). The rationale for selecting the anatomically sterotyped sites at the time of tissue sampling was to ensure maximum coverage of the tumour from superior to inferior; medial to lateral; and anterior to posterior. We chose the oblique fissure (R5) as this is common to both the right and left lung and could be a reliable anatomical site bilaterally. Spatial evolution of mesothelioma would be expected to be best reflected in samples at maximum distance from one another, reflecting the inherent intratumour heterogeneity. These regions are represented by the most superior region (R1) and the inferior (R3 and R4). **B** Inter-patient and intratumour heterogeneity in the MEDUSA22 cohort. Histograms summarising the variance in clonal versus subclonal mutations (left) compared with somatic copy number alterations (SCNAs, right). Left top, relative number of mutations, left bottom, proportion of mutations ranked by clonal mutation proportion ranging from maximum (left) to minimum (right). Right top, ratio of truncal SCNA versus branch SCNA. Right bottom, relative proportion of SCNA ranked as for mutations. **C** Spectrum of subclonal diversity in the MEDUSA22 cohort, ranging from linear topology (MED33 left) to branched (MED12 right). Centre, Pie chart showing the relative proportion of patients in the MEDUSA22 cohort classified as either linear (64%) or branched (36%). **D** Box plots showing the positive correlation between topology classified as either linear or branched, and subclonal SCNA count (two-sided Mann–Whitney U test p = 0.034, left) and subclonal mutation count (two-sided Mann–Whitney U test p = 0.003, right). n = 22 patients. Both box plots denote medians (centre lines), 25th and 75th percentiles (bounds of boxes), and minimum and maximum (whiskers).

hit events were significantly enriched in tumour suppressor genes compared with other genes and agreed with the mesothelioma TCGA. Clonal second hit events most frequently involved *NF2* in 7 MPMs, followed by *BAP1* in 6 MPMs and *SETD2* in 3 MPMs (supplementary fig. 7D). Germline *BAP1* has been reported previously[22] but this was not identified in *BAP1* bi-allelic inactivation in this cohort. Subclonal bi-allelic inactivation commonly involved *BAP1* and the Hippo pathway genes *NF2* (MED34) and *LATS1* (MED3), (supplementary fig. 7D).

To explore the impact of driver evolution on MPM progression, we conducted longitudinal biopsies and copy number analysis of a patient who had exceptionally indolent MPM exhibiting a 12-year progression-free survival interval following radical surgery. Subclonal *NF2* copy number loss was observed at the time of radiological progression, and this coincided with both a higher burden of copy number alterations and chemoresistant MPM, suggesting late stage subclonal expansion, and Hippo pathway inactivation coincident with clinically more aggressive growth and drug resistance (Fig. 2b).

In contrast to the low mutation burden in this cohort, copy number alterations involved almost a quarter of the exome 24.71% (range 12.76–92.58) of which the majority were clonal, 21.32% (1.17–63.22) and 4.91% (0.1–29.36) were subclonal (supplementary

data 12 and 13). Copy number losses, estimated and cross validated using 5 algorithms and by array based analysis (supplementary fig. 8), were significantly more common than gains across the cohort (Fig. 2c, supplementary fig. 9 and supplementary data 12 and 13).

Clonal amplification was observed in three cytobands including 16p11.2 which encompasses the oncogenic transcription factor gene *FUS*[23] commonly involved in translocations associated with myxoid liposarcoma, 17q21.31, and 17q22-23 which encompasses the E3 ubiquitin ligase substrate binding adaptor gene *SPOP*[24,25], most commonly mutated in prostate cancer (supplementary figs. 1 and 10A, B). The most frequent clonal gains involved 8p24 in 3 MPMs (13%). This region spans *NDRG1* which stabilises methyltransferases involved in DNA repair, and also *RECQL4* which plays a crucial role in the end resection step of homologous recombination. Only one patient had 8p24 gain as a subclonal event suggesting that this event is restricted to the earlier phase of MPM evolution.

MPMs showed evidence of allelic heterogeneity, with clonal copy number alterations mirroring positively selected clonal mutations. The most frequent was -22q (*NF2*) in 82% of MPMs, followed by 9p21.3 (*CDKN2A* and *MTAP*), and 3p21.3 (*BAP1*), (supplementary fig. 1 and supplementary data 12 and 13). We

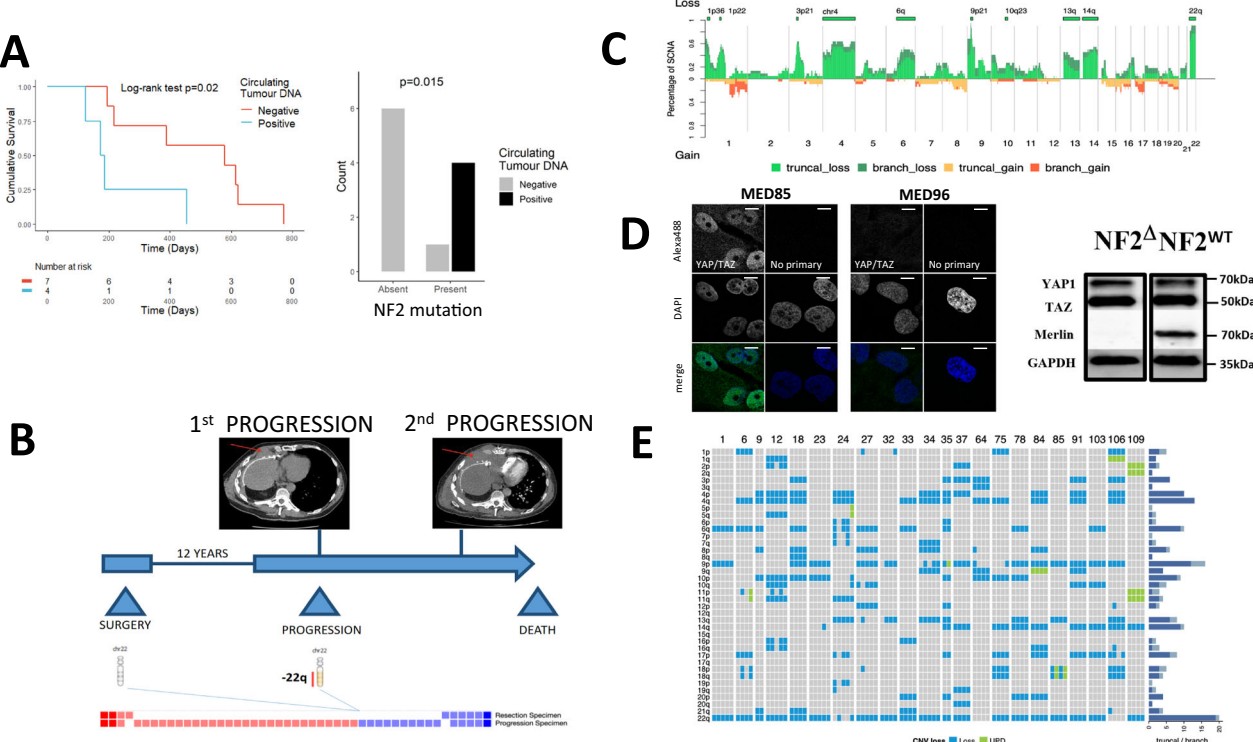

**Fig. 2 NF2/22q loss correlates with detectable ctDNA and late progression in MPM. A** Left. Kaplan-Meier plot showing the relative survival of patients with detectable, clonal driver mutations present in the circulating tumour DNA (ctDNA). Overall survival was significantly lower for patients positive for ctDNA (Two-sided Log-rank test $p = 0.020$; hazard ratio 5.20; 95% confidence intervals: 1.12, 24.09). Right. Column chart showing that patients who had detectable ctDNA (black column) were more likely to have a NF2 mutant tumour (two-sided Fisher's exact test $p = 0.015$; Odds ratio is infinite because all ctDNA-positive tumours were NF2 mutants; 95% confidence intervals: 1.27, infinite). $n = 11$ patients. **B** Longitudinal sampling of tissue from a patient taken at the time of radical pleurectomy decortication (this patient was not enroled into MEDUSA). Array based SCNA analysis was conducted showing subclonal loss of 22q at first progression with a higher SCNA burden, ~12 years following surgery. Subsequent progression was associated with drug resistance (CT scan top right). The squares at the base show copy number losses in blue and gains/amplifications in red. The time from final progression to death was 29 days. **C** Stacked mountain plot summarising clonal versus subclonal losses (above, green) compared with gains (orange/truncal, red/branch) below. This figure shows the relative frequency ($y$ axis) of copy number losses compared with gains along the whole-exome ($x$ axis) averaged across the cohort of patients. **D** Nuclear YAP expression in a cell line derived from patient MED85 harbouring deletion of NF2, compared with wild-type NF2 (MED96) showing non-nuclear YAP expression (right panel). Confocal images: FV1000 (Olympus), ×60 objective, ×4 zoomed scale bars = 10 μm. **E** Heatmap summarising arm level losses in the MEDUSA22 cohort by chromosome (left column). The clonal (blue) versus subclonal (grey) frequency of arm level losses per chromosome are shown in the histogram on the right.

orthogonally validated functional inactivation of the hippo pathway by confocal microscopy in paired exome sequenced primary cell lines grown from MEDUSA MPMs. Clonal *NF2* deletion was present in the cell line MED85 and this was associated with constitutive nuclear translocation of YAP, compared with MED96 which harboured wild-type *NF2* (Fig. 2d). Across the cohort, alternative modes of Hippo pathway inactivation were observed involving *LATS1* (35% clonal versus 11% subclonal), and *LATS2* (32% clonal versus 18% subclonal), (supplementary figs. 1 and 7D).

Clonal copy number events frequently involved arm-level losses more frequently than gains, occurring at higher frequency on chromosome 4p and q, 9p and 22q (Fig. 2e and supplementary data 14). We did not identify whole-genome haploidisation[7] or whole-genome doubling[26] in this cohort. Copy-neutral loss of heterozygosity (CN LOH also referred to as uniparental disomy) involving duplication of an allele with concurrent loss of the other allele, occurred in 12 patients of which 11 events were restricted to chromosome 9 (supplementary fig. 11 and supplementary data 15).

Clonal homozygous deletion involving the region 9p21.3, which harbours the tumour suppressors cyclin-dependent kinase inhibitor 2A and B (*CDKN2A/B*) and methylthioadenosine phosphorylase (*MTAP*), was found in seven patients (32%).

Homozygous deletion only ever occurred as a clonal event, whereas subclonal heterozygous loss was seen in six patients (26%).

With the exception of 9p21.3, clonal homozygous deletions were otherwise rare, involving 3p21 (*BAP1*) in only one patient. Mirrored subclonal allelic imbalance (MSAI) involving spatially distinct, reciprocal copy number losses was a rare event, occurring in only one MPM in 10q23 (*PTEN*) consistent with parallel evolution (supplementary fig. 12).

One MPM (MED109) exhibited no driver single nucleotide variants (SNVs) or INDELs, had more copy number gains than losses, and had clonal deletions involving multiple drivers *CDKN2A* and *NF2*. In addition, this patient also had deletion of *ID3* (1p36.12), and a region spanning 14q11-14q32 harbouring the genes *CCNB1IP1* (14q11.2), *BAZ1A*, *NKX2-1* (14q13), *MAX*, *RAD51B* (14q23-24), *DICER1*, and *BCL1B* as well as -22q encompassing *LZTR1*, *ZNRF3*, *MYH9*, *APOBEC3B*, *MRTFA*, *PATZ1*, *EP300* and *CHEK2* (supplementary fig. 1 and supplementary data 13).

The clonal mutation order associated with early evolution of MPM is unknown. We applied transfer learning[27] to de-noise and structurally correlate trees across the MEDUSA22 cohort (supplementary fig. 13) in order to determine whether MPMs shared

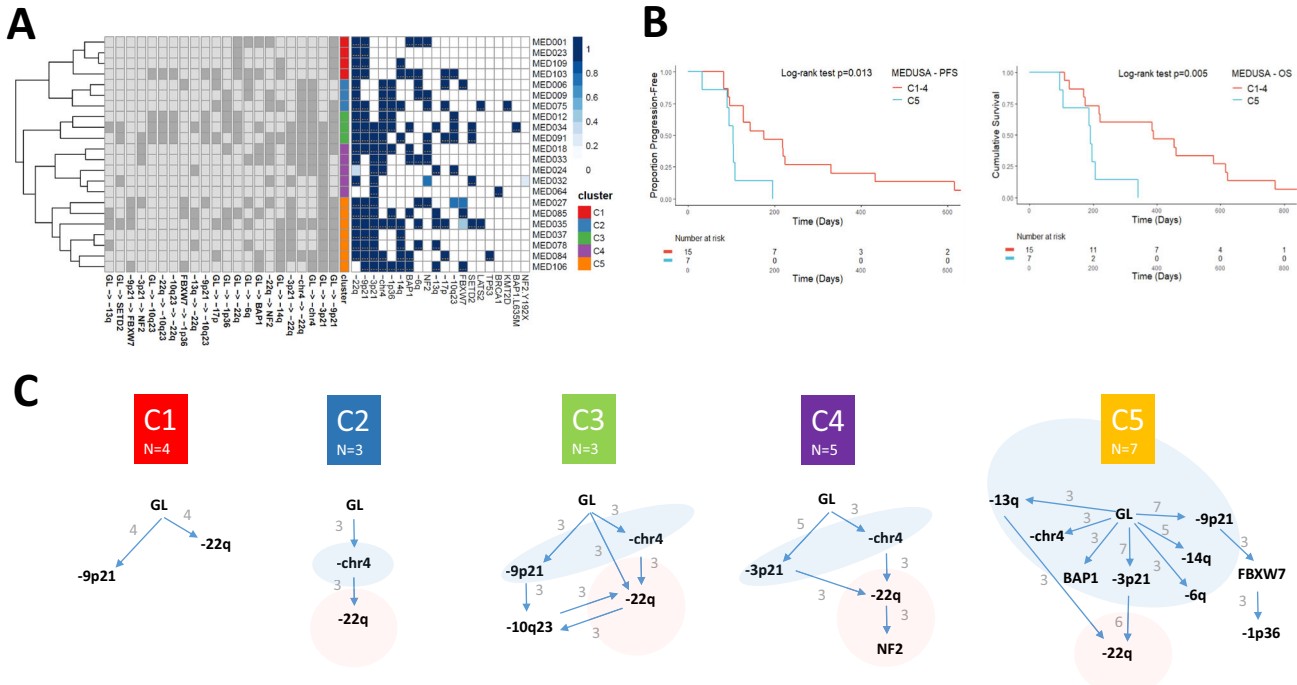

**Fig. 3 Defining evolutionary trajectories and their prognostic significance in MPM. A** Heatmap showing the evolutionary clusters inferred by transfer learning and hierarchical classification. In the middle of the heatmap, five evolutionary clusters are defined and are vertically colour coded, with cluster 1 (C1) in red, C2 blue, C3 green, C4 purple and C5 orange (shown in the key on the right). On the left side of the heatmap, individual evolutionary transitions (i.e. mutational events including both SCNAs and mutations) are shown on the lower axis. The abbreviation GL corresponds to germline. Any transition occurring more than three patients is considered a repeated evolutionary transition. On the right the presence or absence of a mutation or copy number loss is shown. On the right, the probability of an event is presented by the shade of blue, with probability = 1 being dark blue. **B** Kaplan-Meier curves showing (left) shorter progression-free (two-sided Log-rank test $p = 0.013$; hazard ratio 3.52; 95% confidence intervals: 1.22, 10.13) and (right) lower overall survival (two-sided Log-rank test $p = 0.005$; hazard ratio 4.43; 95% confidence intervals: 1.42, 13.77) of C5 MPMs compared to other evolutionary clusters (C1–C4). $n = 22$ patients. **C** Inferred evolutionary trajectories involving repeated transitions in more than three patients. Somatic events that are exclusively early clonal (germline (GL) transitions) are encompassed by a light blue area. Conversely, exclusively late clonal transitions are highlighted in red. The specific number of patients exhibiting repeated transitions is shown as numbers against the directed edges (transitions).

common evolutionary trajectories with a constrained clonal mutation order, and to define critical bottlenecks[28] during evolution that could serve as potential targets for stratified therapy. Integrating both clonal SCNA and SNV driver events, 5 evolutionary clusters were delineated, ranging from low complexity (cluster 1) to the most complex (cluster 5 Fig. 3a, c). A Jackknife procedure was used to estimate MPM evolutionary cluster stability (supplementary fig. 14)[27].

Notably the largest cluster C5, which exhibited the largest number of repeated early clonal transitions, and featured frequent *FBXW7* mutation (supplementary fig. 6) secondary to 9p21 loss, was associated with the shortest median progression free survival (logrank $p = 0.013$) and overall survival (logrank $p = 0.005$ Fig. 3b). We utilised a decision tree to bin MPMs into evolutionary clusters from the tumour genome Atlas, and confirmed a poorer prognosis for C5 (supplementary fig. 15 and supplementary data 16). Notably, in both the MEDUSA22 cohort and TCGA, C5 was restricted entirely to the epithelioid subtype (supplementary data 16).

Allelic heterogeneity associated with either *BAP1* or its locus 3p21 was a repeated, universally early event during MPM evolution involving 2 out of 5 evolutionary clusters (C4 and C5). Cluster C2 was an outlier exhibiting chromosome 4 loss (-chr4) defined by at least 60% loss of the whole chromosome and which encompassed *FBXW7*(4q31.3). This was the only repeated early clonal event in cluster C2, but was also associated with other early clonal events in C3, C4 and C5 suggesting a frequent and critical role for this macroevolutionary event during MPM tumorigenesis.

A consistent feature across the evolutionary clusters was the timing of hippo pathway inactivation signified by either loss of 22q or *NF2* mutation. This was frequently found to be a late clonal event involving all clusters except C1.

The tumour microenvironment was modulated by the degree of exonic ITH with evidence of genomic immune evasion implicating an interplay between host immunity and MPM evolution. Evolutionary cluster C5 was associated with higher CD8 T lymphocyte infiltration (Fig. 4a). The MEDUSA22 cohort showed a wide inter-patient variability in predicted neoantigen burden averaging 15 per patient (range 0–27) of which the majority were clonal, with a median of 12 neoantigens per patient (0–26), compared to subclonal neoantigens which averaged 1/patient (range 0–14; Fig. 4b). Neoantigens were recurrent in 6 genes including two drivers: SETD2 which was the most common (32%), and *LATS2*, as well as *XIRP2*, *TMEM104*, *TTN* and *TME1L1* (supplementary data 17). Subclonal neoantigen burden was associated with lower immunosuppressive regulatory T cell infiltration, as assessed by transcriptome-based digital cytometry (Wilcoxon $p = 0.018$, Fig. 4c).

Tumours achieve immune escape from cytolytic T cell immunosurveillance, in part via loss of heterozygosity (LOH) of the human leucocyte antigen (HLA) cluster, an immunoediting strategy that reduces antigen expression via the major histocompatibility complex (MHC), to escape T cell receptor-dependent CD8 lymphocyte mediated tumour suppression. HLA LOH was found in five patients and was always subclonal (supplementary fig. 17 and supplementary data 18), ie. temporally

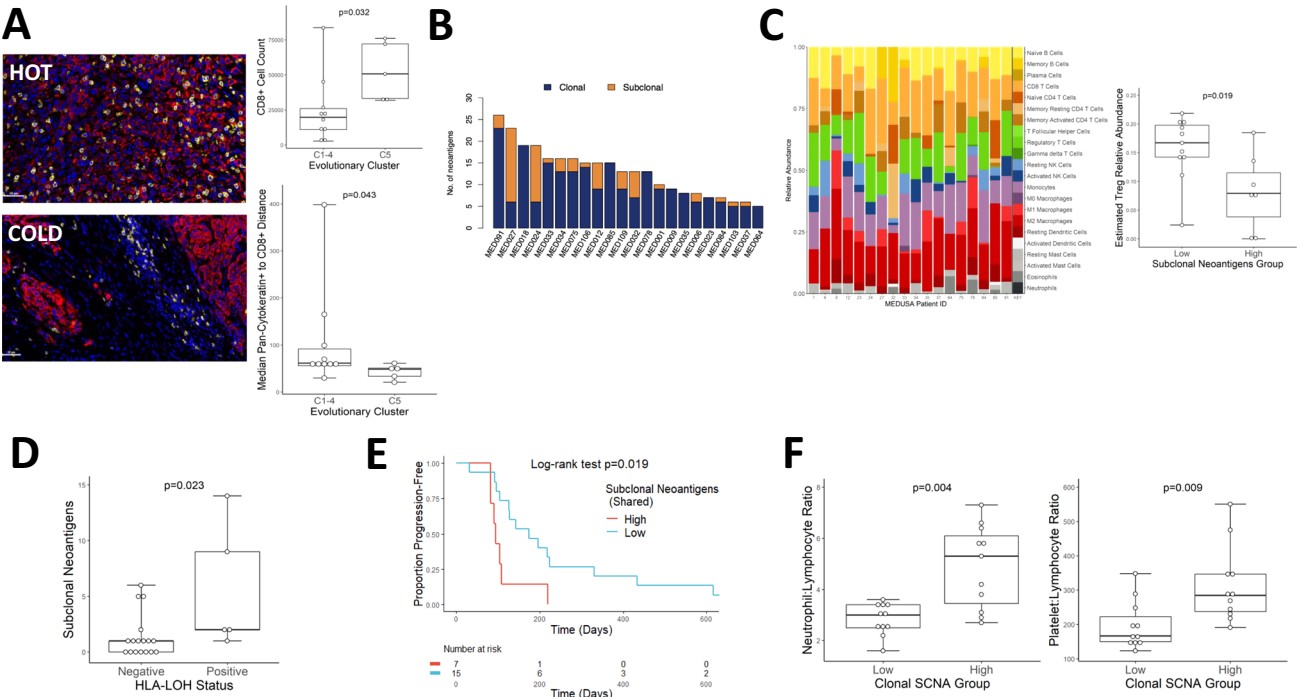

**Fig. 4 Clonal architecture shapes the immune microenvironment in MPM. A** Left. Representative immunofluorescence micrographs contrasting cluster C2 CD8$^+$ exclusion ("COLD", MED75) with a highly infiltrated C5 tumour ("HOT", MED35). Yellow: CD8; Red: Pan-Cytokeratin; Blue: DAPI (nuclei). Original magnification ×400. Scale bars represent 50 µm. Right. Higher CD8$^+$ T-cell infiltration associated with evolutionary cluster C5 by cell count (Two-sided Mann–Whitney $U$ test $p = 0.032$) and Pan-Cytokeratin to CD8 distance (two-sided Mann–Whitney $U$ test p = 0.043). $n = 15$ patients. Both box plots denote medians (centre lines), 25th and 75th percentiles (bounds of boxes), and minimum and maximum (whiskers). **B** Histogram showing the relative number of neoantigens across the MEDUSA22 cohort, colour coded as either clonal (blue) or subclonal (orange) ranked from maximum (left) to minimum (right). **C** Left. Stacked column chart summarising the relative proportion of tumour infiltrating leucocytes enumerated by digital cytometry using CIBERSORT. Right, box plot showing a significantly higher Treg abundance in MPMs with lower neoantigen diversity (two-sided Mann–Whitney $U$ test $p = 0.018$). $n = 18$ patients. Box plot denotes median (centre line), 25th and 75th percentiles (bounds of box), and minimum and maximum (whiskers). **D** Box plot showing increased neoantigen intratumour heterogeneity (ITH) in patients with evidence of HLA LOH (two-sided Mann–Whitney $U$ test $p < 0.023$). $n = 22$ patients. Box plot denotes median (centre line), 25th and 75th percentiles (bounds of box), and minimum and maximum (whiskers). **E** Kaplan-Meier plot showing significantly lower mortality for patients with a higher burden of subclonal (shared) neoantigens (two-sided Log-rank test $p = 0.019$; hazard ratio 3.14; 95% confidence intervals: 1.16, 8.51). $n = 22$ patients. **F** Box plots showing a significant interaction between clonal SCNA burden versus neutrophil-lymphocyte ratio (NLR, left, two-sided Mann–Whitney $U$ test $p = 0.004$) and platelet-lymphocyte ratio (PLR, right, two-sided Mann–Whitney $U$ test $p = 0.009$). $n = 22$ patients. Both box plots denote medians (centre lines), 25th and 75th percentiles (bounds of boxes), and minimum and maximum (whiskers).

restricted to later stages of MPM evolution following exomic diversification. Higher subclonal neoantigen burden was associated with HLA LOH (Wilcoxon $p = 0.023$, Fig. 4d) and time to progression following surgery (Fig. 4e, logrank $p = 0.019$). MPMs harbouring high clonal copy number burden exhibited greater systemic inflammation reflected by Neutrophil:lymphocyte and platelet:lymphocyte ratios (Fig. 4f).

### Discussion

Mesotheliomas are uniquely created following exposure to asbestos[1]. The evolutionary path taken unfolds over several decades to manifest as an enormous cancer that typically grows contiguously to invade local structures, rather than exhibiting distant metastatic behaviour. Our study reveals extensive exomic variation both between and within patients with MPM. This may be reflected clinically in the broad spectrum of MPM phenotypes that extend from highly indolent to aggressive and rapidly proliferative. Epithelioid MPMs are considered the least aggressive histological subtype, compared with sarcomatoid and biphasic MPM. In this study, transfer learning revealed a prognostic cluster C5 which had the most complex evolutionary trajectory and was associated with a shorter survival. In the MEDUSA22 cohort, C5 was entirely epithelioid, consistent with a recent study

that has reported the use of deep learning to identify poor prognostic epithelioid MPMs[29].

Loss of chromosome 4, 3p21 and 9p21.3 were consistently found to be early clonal events, likely involved in initiation. *BAP1* is encoded by 3p21 and is a tumour suppressor gene that encodes a nuclear deubiquitinase that regulates histone H2A to regulate transcriptional repression. *BAP1* mutation or 3p21 loss was predominantly heterozygous, with evidence of clonal and subclonal second hits leading to its complete inactivation. BAP1 has been reported to suppress the polycomb repressive complex PRC2 and its inactivation results in upregulated EZH2 mediated trimthylation of histone H3 (H3K27me3s). Targeting the PRC2 catalytic core (EZH2) with specific small molecule inhibitors may be synthetic lethal to MPMs, as supported pre-clinically[30]. We reported a clonal *BAP1* mutation rate of 36%, similar to that reported in the TCGA[7]. However, minute deletions revealed by single region high density comparative genomic hybridisation array, combined with next-generation sequencing implicates a generally higher rate of *BAP1* somatic alterations in MPM[31].

Our findings suggest that loss of chromosome 4 is a repeated event during early evolution of MPM. *FBXW7* is encoded at 4q31.3 and was identified as a positively selected tumour suppressor. It is recurrently mutated in evolutionary cluster C5.

*FBXW7* is negatively prognostic in univariate analysis in the TCGA and encodes an F box protein family member involved in phosphorylation-dependent ubiquitination . Its inactivation upregulates the prosurvival BCL2 family member, MCL1 leading to resistance to antimicrotubule agents[32,33] such as vinorelbine, which has demonstrated useful clinical efficacy in relapsed MPM[34].

*NF2*/-22q was found to be a predominantly late clonal event, suggesting an early evolutionary constraint. *NF2* mutation was positively selected during both early and late evolution. Parallel evolution involving NF2/-22q implicates a deterministic trajectory in MPM that involves Hippo pathway inactivation, as a key bottleneck during progression of mesothelioma and this is implied in an extreme case whereby *NF2* loss occurred 12 years after surgery at the time of progression. The Hippo pathway is actively being explored as a target for therapy. Although focal adhesion kinase inhibition has been reported to induce synthetic lethality in preclinical models, this has not been borne out clinically[35,36]. However, targeting oncogenic YAP-driven transcription, ferroptosis, or STRN3[37] may present an opportunity to effectively restore Hippo pathway mediated tumour suppression to treat MPM[11,38,39].

Clonal homozygous deletion of *CDKN2A* was identified in approximately a third of the MEDUSA22 cohort. Loss of 9p21.3 carries CDKN2A leading to loss of p16ink4a and derepression of cyclin-dependent kinases 4 and 6 which may serve as a therapeutic target[40]. Accordingly, we have recently completed a clinical trial of abemaciclib to test this hypothesis, which has demonstrated evidence of clinical activity (trials.gov identifier NCT03654833). *MTAP* deletion has been reported to confer sensitivity to inhibition of PRMT5[41,42] or deprivation of its substrate s-adenosylmethionine via inhibition of MAT2A[43], both currently clinical drug development targets with potential in MPM.

We found that MPM evolution shapes the MPM tumour microenvironment. The most genomically unstable evolutionary cluster C5 was associated with higher T-cell infiltration and higher neoantigen burden was associated with immunoediting via HLA LOH, consistent with immune escape, presumably secondary to heightened tumour surveillance. Anti-PDL1 immune checkpoint inhibition has revolutionised the treatment of multiple cancers, with emerging evidence of useful clinical activity in MPM[4,44–46]. Our results suggest that clonal architecture modulates constitutive immune surveillance, antecedent to immune escape involving HLA LOH. This may highlight a potential mechanism leading to de novo or acquired immune checkpoint inhibitor resistance[15] in MPM. Our clinicogenomic correlative studies in clinical trials are underway involving phase II and III[47] cohorts to test this hypothesis.

## Methods

**Ethical and governance approvals**. This research was approved by National ethical committees, under the references 4/LO/1527 (a translational research platform entitled *Predicting Drug and Radiation Sensitivity in Thoracic Cancers – also approved by University Hospitals of Leicester NHS Trust under the reference IRAS131283*) and 14/EM/1159. Sponsor: The University of Leicester).

**Patients and regional tissue sampling**. Patients were approached for enrolment into the MEDUSA cohort if they had a confirmed histological diagnosis of malignant pleural mesothelioma, and were scheduled for routine surgery involving extended pleurectomy decortication at the Glenfield Hospital (University of Leicester). Patients were approached 24 h prior to their operation and provided with patient information regarding the research. All patients signed informed consent prior to surgery. Following surgery, all patients were longitudinally tracked until disease progression with CT monitoring. Progression scans were assessed by a consultant radiologist according to modified RECIST1.1.

During surgery, involving extended pleurectomy decortication (EPD) tumour tissue was sampled consistently from the same anatomical locations:

(1) Apex
(2) Pericardium
(3) Anterior costophrenic recess
(4) Posterior costophrenic recess
(5) Oblique fissure

A 23-blade scalpels were used to cut up to 10 pieces of tumour tissue measuring ~1.5 cm × 0.5 cm ensuring no cross contamination. Eight pieces of tumour were placed on cork and two pieces not on cork. All samples were flash frozen in liquid nitrogen. Cryovials were stored at −80 °C until further use. Hematoxylin and eosin slides were assessed by a histopathologist and tumour content scored as a tumour proportion percentage.

**DNA extraction from frozen tissue and blood**. The extraction of tumour tissue DNA was performed using the QiAamp mini kit (Qiagen), according to the manufacturer's instructions. Four to six sections were used for each region. Prior to each EPD, blood was obtained in five 7.5-ml blood tubes containing EDTA. One of these blood tubes was frozen whole. The remaining four were centrifuged at $226 \times g$ for 10 min at 4 °C. Following this, the plasma was collected and centrifuged at $402 \times g$ for 10 min at 4 °C. The whole blood, buffy coat and plasma were then stored at −80 °C until further use. The extraction of germline DNA was performed using 200 μl of buffy coat and was carried out using the QiAamp DNA blood mini kit (Qiagen), according to the manufacturer's instructions. Qubit DNA High Sensitivity (HS) Assay Kit (Invitrogen) was used to determine quantity DNA. Samples were prepared according to the manufacturer's instructions and read on the Qubit 4 Fluorometer (Invitrogen).

**DNA extraction from cell lines**. The extraction of DNA was performed from $5 \times 10^6$ cells and was carried out using the QiAamp DNA blood mini kit (Qiagen), according to the manufacturer's instructions.

**Multi-region whole-exome sequencing**. Exome capture and amplification. For each tumour region and matched germline, the whole-exome sequencing library was prepared from 1 μg genomic DNA by using the Agilent SureSelect Human All ExonV6 kit (Agilent Technologies, San Diego, CA).Index codes were added to each sample. DNA fragmentation (180-280bp) employed hydrodynamic shearing system (Covaris, Massachusetts, USA). Remaining overhangs were converted into blunt ends via exonuclease/polymerase and enzymes removed. After adenylation of 3′ ends of DNA fragments, adaptor oligonucleotides were ligated. DNA fragments with 3′ and 5′ ligated adaptor molecules were enriched by polymerase chain reaction (PCR), followed by hybriziation with a biotin labelled probe. Streptomycin-coated magnetic beads were then used for exon capture, which were then enriched by PCR to add index tags to prepare for hybrization. Purification employed the AMPure XP system (Beckman Coulter, Beverly, USA). Quantification employed the Agilent high sensitivity DNA assay on the Agilent Bioanalyzer 2100 system.

**DNA sequencing**. Qualified exome capture libraries were then sequenced on Illumina NovaSeq 6000 platform, according to standard protocols, for 150 bp paired-end multiplexed sequence. After sequencing, mean coverage of tumour and normal exomes were both 276X.

**SNV and INDEL calling**

*Sequence alignment*. Raw paired end reads (150bp) in FastQ format were first go through data quality control process. Reads with adaptor contamination, reads containing poly-N, and low-quality reads were removed. All the downstream analyses were based on clean reads that passed quality control. Burrows-Wheeler Aligner (bwa-0.7.17) was used to align the clean reads to human reference genome (UCSC hg19) by the mem algorithm with default parameters.

**BAM file processing**. Mapped genomes were sorted using Sambamba (v0.6.7). We then used Picard tools (v2.18.9) to merge bam files from the same patient region and mark duplicate reads. Statistics was done using SAMtools (v1.8).

**SNV/InDel calling**. Somatic SNVs and INDELs were detected with VarScan2 and MuTect2[12]. Briefly, VarScan2 somatic (v2.3) were used to do somatic variants calling between tumour and matched normal samples based on the output from SAMtools mpileup (1.0)[48]. Default parameters were used except for the following; minimum coverage for normal and tumour sample were set to 10 and 8 separately, minimum variant frequency was adjusted to 0.01 and tumour purity was set to 0.5. After which VarScan2 processSomatic was used to extract somatic variants with minimum tumour frequency 0.01 and maximum normal frequency 0.05. Then bam-readcount (0.8.0) and Varscan2's wrapped fqfilter.pl were combined to conduct the mutation filtering. We used MuTect2 contained in GATK bundle (4.0.5.1) with default parameters.

All detected variants were annotated with Annovar (14 Dec 2015)[49]. Main databases used in future filtering or downstream analysis are as follows: SIFT, PolyPhen and MutationTaster scores used to predict the deleteriousness of

mutations; Alternative allele frequencies in populations reported by large scale sequencing projects 1000 Human Genome(1000G), Exome Aggregation Consortium(ExAC) and exome sequencing project(ESP); Other databases including dbSNP, COSMIC, GO and KEGG.

**SNV/InDel filtering.** To reduce false positive variant calls, further filtering strategies were used on the mutation detection results of both MuTect2 and VarScan2. An SNV would be considered a true positive call if it satisfied the following conditions:

*Variant allele frequency.* The SNV was called by MuTect2 and VarScan2 (somatic *p*-value ≤ 0.1) simultaneously, and both with a variant allele frequency (VAF) no <2%; or only detected by VarScan2, but with a VAF greater than 5%.

*Variant allele frequency in normal samples.* VAF in matched normal samples for the position need to be <1% and reads number for alternative alleles is <5.

*Blacklist filter.* The SNV was not located in the blacklist, which related to specific genomic regions like simple repeats and segmental duplications.

*Population frequency-based filter.* The population frequency of the SNV did not exceed 1% in any of the following population based database- 1000G, EXAC or ESP6500, according to the annotation results of Annovar.

*Force calling.* Multi-regional sequencing allowed the opportunity to increase the sensitivity to detect variants with low frequency. For a somatic mutation that was not detected ubiquitously across all tumour regions in a patient, reads information was extracted from corresponding bam files of each region which with a negative call, by using bam-readcount (0.8.0)[12]. In such cases, if mapping quality >20 and VAF >2%, this site was treated as a positive call.

**Indel filtering.** For InDel filtering, the blacklist filter and population frequency-based filter were the same as described as above. VAF threshold was set to 5% for VarScan2 with somatic *p*-value ≤ 0.05, total read depth was set >50, alternative read depth >10 in tumour samples and <2 in corresponding normal samples.

**Sample contamination using CONTEST.** ContEst (1.0) was used to evaluate cross-individual contamination for all sequenced samples with default parameter. As described [2], ContEst is a GATK module for estimating the level of cross-sample contamination in next-generation sequencing data based on Bayesian framework. The input files used were each sample's bam-file after alignment and corresponding germline SNV mutation result in vcf format detected by GATK as described above. The population allele frequency file used during the analysis was downloaded from the CGA official website along with the ContEst software (http://software.broadinstitute.org/cancer/cga/contest_download).

**SCNA calling**

*Absolute.* We used multiple methods to assess putative cellularity and ploidy. Firstly, CNVkit was performed to generate segmented logR calls for tumour-normal pairs, and ABSOLUTE was used to estimate tumour purity, ploidy and absolute copy number per segment. The ABSOLUTE solutions were reviewed manually according to the recommended best practice. Beside, as the majority of ubiquitous mutations are likely clonal, their VAFs (after correcting for copy number) should reflect the tumour purity, says the maximum peak of variant allele fraction distribution should equal to one half of the purity. ASCAT/Battenberg estimates of purity and ploidy were also included to compare with ABSOLUTE solution. Therefore, we assessed whether the tumour purity and ploidy esimates from copy number analysis (ABSOLUTE/ASCAT/ Battenberg) were consistent with VAF estimates among multi-regions. We excluded samples with low purity and the VAF peak was <0.1, or samples had no concordant solution.

*ASCAT.* To estimate somatic copy number alternations, ASCAT/Battenberg was performed on multi-regional paired tumour-normal sequencing data. Allele counts of positions from 1000 genomes were generated using AlleleCounter, and minimum coverage of 20 for normal sample was used for filtration. LogR and BAF values were produced for each region, and concatenated into one matrix separately for each patient. LogR values were subsequently corrected using a GC wave correction implemented in ASCAT, and only heterozygous BAF values were reserved for further analysis. Allele-specific multi-sample segmentation was performed to generate segmented logR and BAF data by ascat.asmultipcf. Both ASCAT and Battenberg algorithms were applied to provide cellularity (purity) and ploidy estimates with gamma setting of 1, and manual verification was used to select the optimal model for ploidy and cellularity using an orthogonal measures based on ABSOLUTE results and mutation variant allele fraction. And then ASCAT/Battenberg was re-run to obtain the final allele-specific copy number data using reviewed cellularity and ploidy. To assess SCNA calls, we compared absolute copy number profiles using four software packages – ABSOLUTE, CNVKit, ASCAT and Battenberg. All the samples showed high concordance except patient 12 R1 and R4, which had no best solution in ABSOLUTE.

**Phylogenetics.** We used ASCAT (purity, VAF and SCNA) to obtain the cancer cell fraction (CCF). For each mutation, $n_{mut}$ was calculated. This variable is the product of the CCF for specific mutation and the corresponding copy number (1):

$$n_{mut} = VAF\frac{1}{p}\left[pCN_t + CN_n(1-p)\right] \qquad (1)$$

VAF is the variant allele frequency, and $p$ is tumour purity, $CN_t$ is the tumour locus- specific copy number, $CN_n$ is the normal locus specific copy number ($CN_n = 2$). Next, nchr (expected mutation copy number) was calculated using the VAF, assigning a mutation to one of the possible local integer copy number states, using maximum likelihood.

**PyClone Dirichlet process clustering.** Briefly, for each mutation, the observed variant count was used with a reference corresponding to VAF = 0.5x pre-clustering CCF, major allele copy number = 2, minor allele copy number = 0 and purity = 0.5; Clonal versus subclonal mutations were grouped according to pre-clustering CCF. PyClone was run with the following parameters; 10,000 iterations, burn-in = 1000, --var_prior = 'BB' and –ref_prior = 'normal'.

**Phylogenetic trees.** Clonal deconvolution in Pyclone served as an input to generate phylogenetic trees in CITUP (CITUP, v0.1.1, https://github.com/amcpherson/citup)[50]. Default parameters were used for the QIP mode, mutation clusters and mean CCF as inputs. Mutations< per cluster were filtered and trees ranked by BIC (Bayesian information) score, with the tree solution being selected at the minimum. clonevol package[51] was used to profile the subclonal architecture in each tumour region, and trees were drawn using igraph.

**Evolutionary cluster inference by transfer learning.** REVOLVER was used to infer repeated evolutionary trajectories in MPM[27]. This package uses the pigeon-hole principle and sum rule for phylogenetic construction, and implements a maximum likelihood (ML) method to jointly fit and identify similar trajectories among patients by transfer learning. Evolutionary distances between fitted trajectories were computed and used to stratify the cohort into subgroups of tumours that harbour similar evolutionary trajectories. We jointly considered both somatic mutations and copy number alternations occurring in NF2, BAP1, FBXW7, SETD2, LAST2, PTEN and CDKN2A as drivers in MPM. CCF values of mutations were obtained from PyClone analysis. While SCNAs were assigned to CCF clusters by present or absent in multi-regions, their CCFs were defined as the mean CCFs of target cluster. A jackknife approach was applied to estimate the stability of clusters and trajectories. REVOLVER analysis was conducted in R version 3.5.1.

Prognostic impact of evolutionary clusters was orthogonally validated using the mesothelioma TCGA dataset[27]. Briefly, a decision tree was used to bin MPMs in the TCGA dataset. These clusters were then compared for overall survival using Kaplan-Meier analysis.

**Driver calling.** Driver genes were identified combining multiple strategies. Firstly, dNdScv (https://github.com/im3sanger/dndscv) and MutsigCV(v1.4) were used to predict driver genes separately based on somatic mutations detected above for the whole cohort under analysis, both with default parameters. Genes with *q*-value < 0.05 reported by either methods were considered as candidate driver genes. All non-silent variants were compared with a potential driver genes list (*n* = 26), curated from previous pleural mesothelioma analyses[7,52]. Any variants with location overlap with one of these genes were selected. The newly predicted as well as reported genes in the literature, comprised our candidate driver gene list. Then, functional prediction was done to infer whether these candidate driver gene associated variants were deleterious or not. Four computational approaches were applied-SIFT, Polyphen (Polyphen2_HDIV and Ployphen2_HVAR), MutationTaster and CScape (http://www.cscape.biocompute.org.uk). A mutation was predicted deleterious if results of both CScape and MutationTaster were deleterious, or CScape with low-confidence deleterious but other 3 algorithms were predicted deleterious.

**dN/dS analysis.** The dN/dS represents ratio of substitution rates at non-synonymous sites to those at synonymous sites, estimating positive selection at mutational level. The dN/dS ratios in this analysis were estimated using the R package dNdScv (https://github.com/im3sanger/dndscv). We separately ran this tool on clonal mutation set, subclonal mutation set, as well as total mutations detected above, with default and recommended parameters.

**Mutation signature analysis (v3).** COSMIC mutational signatures v3 released 59 Single Base Substitution (SBS) Signatures, which can be used to decipher mutational processes in most cancers, such as DNA mismatch repair deficiency, tobacco smoking, UV-light exposures and drug treatment. We used R package deconstructSigs to reconstruct the mutational profiles of each sample based on COSMIC SBS signatures, which applied a multiple linear regression model to determine the

activities of pre-defined signatures. For both clonal and subclonal signature analysis, we excluded samples with mutations less than 50.

**HLA typing and HLALOH**. HLA typing for MHC class-I genes was carried out using POLYSOLVER(v1.0) [3] software for all 28 normal-samples' bam files, with default parameters. In brief, reads in the WES data potentially originate from HLA gene region were extracted out and then aligned to genomic sequence library of all known HLA alleles based on IMGT, using Novoalign packaged in POLYSOLVER. After which, a two-step Bayesian classification approach was used to infer the two alleles for each HLA class-I genes (HLA-A, HLA-B and HLA-C). A crucial part of neoantigen presentation is the HLA class-I genes products, which can present tumour associated epitopes to T cells and then trigger out immunal response of body. Loss of heterozygosity in HLA genes may lead to decreased ability to present productive tumour neoantigens, which could facilitate immune evasion of cancer. LOHHLA[15] (https://github.com/slagtermaarten/LOHHLA) software was used to evaluate HLA loss for all 118 tumour samples, based on the alignment results of both tumour and corresponding normal samples, inferred tumour purity and ploidy information, and the HLA class-I genotyping results detected above. In brief, HLA reads were extracted and re-aligned to the patient-specific HLA-I alleles, then HLA gene specific log ratio was calculated based on coverage information on mismatch positions between homologous HLA alleles, and finally, HLA haplotype specific copy number was determined. In the analysis, items with PVal_unique ≤0.01 (difference in log ratio between allele 1 and allele 2 ≤0.01) were considered as a LOH event.

**Neoantigen prediction**. In this analysis, neoantigens were defined as 8-11-mer peptides resulted from somatic SNVs or InDels which lead to amino-acid changes and, binding affinity score between remodelled peptide and respective patient's HLA class-I molecules was <500 nM. Somatic mutation VCF files both from VarScan2 and Mutect2 were annotated by Variant Effect Predictor (Version 84) with default parameter, except for the using of 'downstream' and 'wild-type' plugins offer by pVACseq[53]. After annotation, the variants items lead to peptide changes were extracted out for downstream analysis. Bam-readcount (0.8.0) was used to acquire sequencing-based read depth information on each selected variant for both tumour and matched normal samples. Annotated non-synonymous mutations, sequencing-based information as well as HLA class-I gene typing results inferred by POLYSOLVER were feed into pVACseq(4.0.9) for neoantigen prediction. For each pVACseq run, epitope prediction was done by both NetMHC and NetMHCpan algorithms packed in pVACseq toolkit, epitope length was set to 8–11 and tumour DNA VAF cutoff was set to 10, with default parameters used for all other settings. Epitope prediction was performed based on the selected prediction algorithms, after which, sequencing-based information was integrated to enable filtering of neoantigen candidates (Normal Coverage ≥5X, Normal VAF ≤2%, Tumour Coverage ≥10X, Tumour VAF ≥10%). Inferred neoantigen candidates were selected out and those with binding affinity fold change >2 were considered with higher priority level, which means the ratio of binding affinity score between wild-type peptide and mutated peptide. The greater this value, the stronger of the binding affinity after mutation compared with wild-type epitope.

**RNA extraction from FFPE**. RNA was extracted 30 formalin-fixed paraffin-embedded (FFPE) tissue blocks. Tumour content was assessed by a pathology. Only samples with more than 50% tumour content were used for the analysis. RNA extraction was carried out with the Quiagen RNeasy FFPE kit, according to the manufacturer's instructions. Four (4) 10-µm-thick sections from each of the FFPE tissues blocks were used. Qubit RNA High Sensitivity (HS) Assay Kit and Qubit 4 RNA Integrity and Quality (IQ) Starter Kit (Invitrogen) were used to determine quantity and quality of RNA, respectively. All samples had an IQ > 5. Samples were prepared according to the manufacturer's instructions and read on the Qubit 4 Fluorometer (Invitrogen).

**Transcriptome analysis and digital cytometry**. Transcriptomic analysis was performed by Hologic Ltd. Samples (1.7–5.5 µg RNA) were analysed on Human HT Clariom™ S pico array, (Thermo Fisher) according to the manufacturer's instructions. Cel. Files obtained from Clariom S analysis were analysed with CIBERSORT, a publicly-available deconvolution algorithm that can identify cell populations and estimate their relative abundances from gene expression data obtained from bulk tissues (Newman et al. 2015). Algorithm execution and stacked bar chart creation were performed on the CIBERSORT website: https://cibersort.stanford.edu/. Resulting figures produced, in R Studio version 1.1.463, running R version 3.5.2.

**Multiplex immunofluorescence based cytometry**. Formalin-fixed, paraffin-embedded (FFPE) sections were cut from MEDUSA22 cohort archival blocks and subjected to deparaffinisation in an alcohol gradient before rehydration and washing. Next, endogenous epitopes were exposed by microwaving samples in 1% Tris-EDTA (pH 9.0) at full power for 20 min. Target-specific staining was achieved using the Opal fluorescent IHC kit (Akoya Biosciences): Opal protein block was added to sections before primary antibody incubation to minimise non-specific staining. Primary antibodies targeted human epitopes of: CD68, clone PG-M1

(diluted 1:500); CD8, clone C8/144B (diluted 1:400); and Pan-cytokeratin, clone MNF116 (diluted 1:200). All primaries were raised in mice, acquired from Dako and diluted in PBS. Standard Opal HRP-conjugated secondary antibody polymer enabled detection of specific primary antibody binding to the target epitope. Fluorophores used were Opal 520, Opal 570 and Opal 690, respective to primaries mentioned above. All stained and no-primary control slides received a DAPI counterstain. Further, unstained sections were prepared as described above, but without antibody or fluorophore incubation, and used as templates of endogenous auto-fluorescence to enable more accurate assessment of fluorophore intensity.

Images were obtained using the Vectra Polaris™ multispectral fluorescence scanner (Akoya Biosciences). The accompanying software, inForm® (Akoya Biosciences), used a machine learning approach to identify nuclei according to DAPI intensity and to phenotype cells. During phenotyping, inForm considered the fluorophore intensity of user-defined positive cells and applied the appropriate phenotype to unsupervised cells with a similar signal. The R programming environment (version 3.6.1, manipulated via R Studio 1.2.5001) was used to measure absolute stained or counterstained cell quantities and compute nearest neighbours of each phenotype per cell.

**Primary cell lines**. MED85 (NF2 deleted) and MED96 cells were isolated from tissue acquired at surgery. Tissue was manually minced on a wax sheet using a scalpel, placed in HBSS media and incubated for 30 min at room temperature. Cells were filtered through 100-µm cell strainer and centrifuged at RT, $300 \times g$ for 10 min. The pellet was resuspended in red blood cell lysis buffer and incubated for 10 min. After centrifugation, cell lines were grown with the presence of 10% CO2 on RPMI medium (31870, Gibco) supplemented with 10% FBS (F9665, Sigma-Aldrich), 1x GlutaMax (35050038, Gibco) and 1% Penicillin/Streptomycin (11548876, Gibco). When about 80% confluent, cells have been harvested using 1x trypsin EDTA solution (15400054, Gibco) and manually counted. Cells were resuspended in phosphate-buffered saline (10209252, Oxoid) with 0.5% of bovine serum albumin (A7030, Sigma-Aldrich) and aliquoted $1 \times 10^5$ per tube. Prepared aliquots were incubated for 30 min at 4 °C with anti-human mesothelin phycoerythrin conjugated antibody (R&D systems) and/or anti-podoplanin – APC conjugated antibody (Miltenyi biotec), or as an unstained control. Dilution of both antibodies was 1:10. After incubation samples were washed three times in PBS with 0.5% BSA (5 min at 300xG) and resuspended to final volume of 300 µl. Samples were acquired on a BD FACS CantoII flow cytometer running BD FACSDiva8.0.1 software, directly after preparation. Cell population abundance: MED85 7500 cells with 25.9% positive for mesothelin and 97.7% positive for podoplanin. MED96 10000 cells with 17.4% positive for mesothelin and 55.7% positive for podoplanin.

Gating strategy: samples were acquired first on FSC vs SSC and a generous gate created. Threshold for APC and PE channels were set on unstained controls. Cell lines were whole-exome sequenced to verify somatic SNV and SCNA profiles compared with the parental tumours.

**Confocal immunofluorescence imaging**. MED (MIST) 85 and MED (MIST) 96 cells grown at low densities on sterile glass coverslips were fixed in 4% paraform aldehyde for 10 min, permeabilized with 0.4% Triton-X100 in PBS for 15 min, and blocked in 5% BSA for 30 min. Primary antibody staining was performed with anti-YAP1 antibody (Santa Cruz Biotechnology), for 1 h at room temperature, followed by secondary staining with chicken anti-mouse IgG (H+L)-AlexaFluor488 (Thermo Fisher Scientific, 1:1000) for 30 min. Cells were counterstained with DAPI and mounted with ProLong Gold anti-fade reagent (Thermo Fisher Scientific). Confocal imaging was performed using an Olympus FV1000 confocal laser scanning system with an inverted IX81 motorised microscope equipped with UPlanSApo 60×/1.35NA objective (Olympus). Obtained images were deconvoluted using Huygens Essential software (Scientific Volume Imaging) and processed using ImageJ software (NIH).

**Clinico-pathological variables**. Clinical data included related to the date age, gender, smoking history, asbestos exposure, histological subtype, TMN stage, receipt of neoadjuvant therapy, laterality of pleural mesothelioma, resection margin and blood parameters: haemoglobin, platelets, neutrophil and lymphocyte counts, CRP and LDH. EORTC prognostic score, Neutrophil-to-lymphocyte and platelet-to-lymphocyte ratios were calculated.

**Isolation of plasma and buffy coat from whole blood**. Blood samples were collected in EDTA tubes and blood components (buffy coat and plasma) were isolated ≤2 h post-venepuncture. Whole blood was firstly centrifuged for 10 min at $1000 \times g$ at 4 °C to separate blood components. Plasma was carefully transferred to clean plasticware, avoiding visible contamination from the buffy coat and packed erythrocytes, and then centrifuged for a further 10 min at $2000 \times g$ at 4 °C to exclude any residual cellular matter, before finally being aliquoted into 1.5 mL tubes. Both centrifugation steps were conducted with the centrifuge brake set to zero. Leucocytes, a source of germline control DNA, were isolated by carefully pipetting the buffy coat layer and transferred to 1.5 mL tubes. Both plasma and buffy coat aliquots were placed at −80 °C for later use.

**cfDNA extraction from plasma.** cfDNA was isolated from 3 mL plasma using the QIAamp® Circulating Nucleic Acid Kit (Qiagen) and the QIAvac 24 Plus vacuum manifold system, according to the manufacturer's protocol. This kit uses a silica membrane-based DNA purification technology, where using several washing steps, contaminants (e.g. proteins, RNA) are effectively removed. Each sample was eluted in a total volume of 150 μL Buffer AVE and then quantified (see, 'Quantitation of cfDNA'). Quantified cfDNA was subsequently stored at 4 °C for immediate use (up to 6 weeks) or at −20 °C for later use.

**Quantitation of cfDNA.** Freshly eluted cfDNA was quantified using the 4200 TapeStation System with High Sensitivity D5000 Reagents and ScreenTape (Agilent Technologies), according to the manufacturer's instructions. In addition to quantitation values, the TapeStation system provides a fragment size distribution for each sample. Considering the typical size range of cfDNA to be found between 60 and 300 bp, a region analysis using these boundaries was conducted and this DNA concentration value was taken forward for subsequent droplet digital PCR (ddPCR).

**Clonal variant selection criteria for cfDNA analysis.** The identification of clonal variants for assay design was determined using the following criteria: (1) absent from germline DNA, (2) found in all tumour DNA regions, (3) Tier 1 and Tier 2 predicted drivers (Cancer Genome Interpreter) were prioritised and (4) highest mean VAF amongst tumour regions. If initially selected variants failed QC (see 'cfDNA assay design and validation'), the criteria were relaxed when required.

**cfDNA assay design, validation and optimisation.** Oligonucleotides were designed using the Primer3 tool (Whitehead Institute, MIT). To confirm oligonucleotide specificity, sequences of interest were aligned to the human genome using the NCBI BLASTN tool. Primer/probe candidate sequences were assessed using OligoAnalyzer 3.1 (Integrated DNA Technologies) software, to assess specificity and to predict their propensity to form hairpins and homodimers/heterodimers. Oligonucleotides with a secondary structure $\Delta G$ value $<$−6 kcal/mol were excluded. Amplicon length was verified using the UCSC in silico PCR tool using the GRCh37/hg19 reference genome. Where possible, the following parameters were adhered to: GC content of 40–60%, length of 15–25 bp, primer $T_m$ of 55–62 °C, probe $T_m$ of 65–72 °C, $\leq$3 consecutive G/C bases, a GC-clamp of $\leq$3 G/C bases and an amplicon length of <100 bp. Desalted primers were delivered lyophilised and were subsequently reconstituted using sterile TE Buffer (Invitrogen) to a stock concentration of 200 μM.

Primers were validated using Fast SYBR® Green melt curve analysis (qPCR), prior to the introduction of the probe, in order to assess primer specificity, efficiency and potential secondary structure formation. Each primer was diluted to a working stock of 10 μM and a standard curve was constructed using serially dilutions (1:2 dilution; 7 points) Human Genomic DNA (Roche) ranging from 20 to 0.3125 ng per well. Each of the serial concentrations were run in triplicate, with a no template control (NTC). A total reaction volume of 10 μL comprised 5 μL Fast SYBR® Green Master Mix (2X) (Applied Biosystems), 0.6μL forward primer (600nM final concentration), 0.6 μL reverse primer (600 nM final concentration), 0.2 μL sterile TE Buffer (Invitrogen) and 3.6 μL template DNA. Reactions were set up in MicroAmp® Fast Optical 96-well reaction plates (Applied Biosystems), sealed with MicroAmp® adhesive film (Applied Biosystems), centrifuged and run on the StepOnePlus™ Real-Time PCR System (Applied Biosystems) under the following conditions; 95 °C for 20 s, 50 cycles of 95 °C for 3 s and 60 °C for 30 s and finally a dissociation (melt) curve step. An efficiency value of 90–110% and an $R2 > 0.98$ was considered acceptable. Secondary products, where applicable were identified through the inspection of the melt curve and those identified to have significant non-specific products were redesigned as required.

Assays were optimised by ddPCR through a temperature gradient (53–63 °C) in order to identify the optimum annealing temperature (Ta) of each assay. Optimal Ta was defined by the temperature which allowed the largest discrimination between positive and negative droplets, whilst avoiding non-specific amplification.

**Droplet digital PCR.** Inhouse designed assays each included common forward and reverse primers, with allele-specific (mutant and wild type) probes (see supplementary data 11, for oligonucleotide sequences). In all, 20 μL individual reaction mixes contained the following: 10 μL ddPCR Supermix for Probes (No dUTP) (Bio-Rad), 1 μL mutation assay (20X) and 9 μL of template DNA/H$_2$O. For tumour and germline DNA, 10 ng was loaded into each reaction. For cfDNA, a higher quantity of 20 ng was loaded as a means of achieving a lower limit of detection (LOD). Each 20 μL reaction was loaded into a DG8™ Cartridge (Bio-Rad) along with 70 μL Droplet Generation Oil for Probes (Bio-Rad), then sealed using a DG8™ gasket (Bio-Rad). Sealed cartridges were then placed into the QX200™ Droplet Generator (Bio-Rad), to allow the generation of >10,000 nanolitre sized emulsion droplets, according to the manufacturer's instructions. Following this, generated droplets were carefully transferred to Twin-Tec 96-Well × 250 μL Semi-Skirted PCR plates (Eppendorf) and sealed with a Pierceable Foil Heat Seal (Bio-Rad) at 180 °C using the PX1™ PCR Plate Sealer (Bio-Rad). The sealed plates then underwent PCR in the C1000 Touch™ Thermal Cycler (Bio-Rad) with the following conditions; 95 °C for 10 min, 40 cycles of 94 °C for

30 s and Ta (assay dependent, range 54 °C–63 °C) for 1 min and then 98 °C for 10 min, before cooling to 4 °C. Post-PCR, plates were analysed on the QX200™ Droplet Reader (Bio-Rad) using the QuantaSoft™ Software v1.7 (Bio-Rad) on the Rare Event Detection settings for mutation detection. The number of positive droplets for mutant and wild type (WT) were used to calculate the VAF of respective mutations.

**Statistical analysis.** All statistical analyses involving clinical, immune and clonal architecture characteristics were performed in R Studio, version 1.2.5001, running R version 3.6.1. Due to the relatively small patient cohort, continuous variables were tested for normality using Shapiro-Wilk's test ('shapiro.test' in base R). Only half of all continuous variables were normally distributed according to Shapiro-Wilk's test, so correlations were investigated using Spearman's rank correlation test ('rcorr' in the package 'Hmisc'). Differences in medians (for skewed distributions) were uncovered using the Wilcoxon rank-sum test ('wilcox.test' in base R), while means were compared (where both populations relating to a category obeyed a normal distribution) using Welch's t-test ('t.test' in base R with 'var.equal'=-FALSE) because samples were not paired and typically of different sizes. Similarly, multi-categorical medians were compared using the Kruskal-Wallis rank-sum test ('kruskal.test' in base R). For categorical variables, statistical significance was interrogated using Fisher's exact test due to frequent small group sizes ('fisher.test' in base R). All derivative figures were generated using the package 'ggplot2'. Survival analysis was conducted using the log-rank test (built into 'ggsurvplot' in the package 'survminer', which was also used to plot survival curves), with Kaplan-Meier curves created using 'survfit' from the package 'survival'.

For ctDNA analysis GraphPad Prism 7 (GraphPad Prism Software Inc., CA, USA) was used for statistics tests including Wilcoxon rank-sum test (Mann–Whitney U test). Any P values were considered significant if <0.05.

**Reporting summary.** Further information on research design is available in the Nature Research Reporting Summary linked to this article.

## Data availability

The main data supporting the results are available within this article and its supplementary information.

The raw sequencing is available in SRA Run Selector via [https://www.ncbi.nlm.nih.gov/sra/?term=PRJNA649889] or [https://www.ncbi.nlm.nih.gov/sra/?term=SRP276753] which is hosted by the national Centre for Biotechnology Information, under accession number PRJNA649889. The data is searchable in Entrez via [https://www.ncbi.nlm.nih.gov/sra/?term=PRJNA649889]. All of the other data supporting the findings of this study are available within the article and its supplementary information files and from the corresponding author upon reasonable request.

## Code availability

Scripts related to clonality analysis, mutational signatures, dn/dS analysis and REVOLVER can be accessed via [https://github.com/zanmer/Mesothelioma_evolution_deciphering_drugable_somatic_alterations], DOI [https://doi.org/10.5281/zenodo.4305392]

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

## Acknowledgements

British Lung Foundation- Mesothelioma UK grant RM38G0071, Cancer Research UK Research grant C61811/A24218, Hope against Cancer research grant, A.J.S. was supported by a Royal College of Surgeons fellowship. A.D.G. was funded by an NIHR Academic Clinical Fellowship and a CRUK research bursary, A.B. was funded by a British Lung Foundation studentship M16-9.

## Author contributions

Initiation and study design by D.A.F. Bioinformatics by M.Z., Q.S., J.-L.L., C.S., G.W., P.P.Z., H.Y. and L.B. Subject enrolment, sample, clinical data collection, and review by D.A.F., A.D.G., A.J.S., A.N., D.W., J.D. and A.Bzura. Circulating DNA analysis by L.M. and J.A.S. Histopathology by C.R., P.W.J., A.G., C.P., M.T.S. and J.L.Q. Radiology by A.Bajaj. Immunofluorescence microscopy by J.H., C.P., T.K. and E.Y.B. Primary cell culture by E.Y.B. and S.B. Gene expression analysis by S.B. and N.K. Nucleic acid extraction and quality assurance, ddPCR by M.J., E.J.H., C.P., N.K., S.B., A.G.D. and A.J.S. Statistical support by G.G. Writing of the first draft of the manuscript by D.A.F., M.Z., J.H., E.J.H. and F.D. All authors contributed to the writing and editing of the manuscript.

## Competing interests

The authors declare no competing interests.
