## [Peer Review File · Nature Communications]

REVIEWERS' COMMENTS

Reviewer #1 (Remarks to the Author):

Dr. Fennell and colleagues have submitted this highly significant and elegant body of work that explores the clonal architecture. They found this to be prognostic and play a role in shaping the tumor microenvironment. The key findings are that the evolution of clones are highly variable and cover the spectrum of linear to highly branched. Interestingly, BAP1 and FBXW7 events are early while NF2 events are late. This work is highly original and challenges some long-held and possibly mistaken assumptions about this disease. Work like this helps propel discovery and drug development in mesothelioma which has lagged far behind other disease. The data presented herein are robust and the approach is logical. The conclusions proposed are supported by the data presented. The analyses are robust and appear valid. The key references are included. No further work is requested as my prior comments have been adequately addressed.

Reviewer #2 (Remarks to the Author):

The authors conducted multi-regional exome sequencing of 90 tumor biopsies from 22 malignant pleural mesothelioma patients, which includes 17 with an epithelioid and 5 with biphasic histology. They found that the most frequent mutations occurred in the BAP1 gene and they established that BAP1 and FBXW7 mutations were early driver mutations. NF2 mutations leading to Hippo pathway inactivation instead were late events. They found that very late NF2 mutations occurred in one patient. CDKN2A mutations were also relatively frequent. Instead, other gene mutations were not frequent. Overall the number of mutations was quite low, and copy number changes were relatively high. The study is quite important for the mesothelioma field although the cohort size is relatively small.

The authors addressed most of the questions in this revised manuscript. However, there is still an important issue that needs to be addressed in order to make an accurate conclusion. It is regarding the frequency of BAP1 mutation found in the somatic mesothelioma specimens. Although the authors stated that their reported frequency of 36% is similar to what was reported in the TCGA. And in the rebuttal letter, the authors also mentioned another two papers (Bueno et al *Nature Genetics* 2016, and Bott et al, *Nature Genetics* 2016-which in fact should be 2011) that had found similar results. Unfortunately, if the authors checked more carefully and thoroughly, they will find many more recent publications using comprehensive methods to check somatic alterations of BAP1 in mesothelioma tumor specimens, and the results from many research groups all confirm about 60% somatic alterations, for example, Nasu M et al *J Thorac Oncol* 2015. Yoshikawa Y et al. *PNAS* 2016;

Lo Iacono M et al, J Thorac Oncol 2015; Hmeljak J et al, Cancer Discovery 2018. Therefore, the authors should definitely think carefully about how to address this issue and make a more accurate statement.

Reviewer #3 (Remarks to the Author):

The authors have now satisfactorily addressed my comments. I have no further comments about this work. Good luck with publication.

REVIEWERS' COMMENTS

Reviewer #1 (Remarks to the Author):

Dr. Fennell and colleagues have submitted this highly significant and elegant body of work that explores the clonal architecture. They found this to be prognostic and play a role in shaping the tumor microenvironment. The key findings are that the evolution of clones are highly variable and cover the spectrum of linear to highly branched. Interestingly, BAP1 and FBXW7 events are early while NF2 events are late. This work is highly original and challenges some long-held and possibly mistaken assumptions about this disease. Work like this helps propel discovery and drug development in mesothelioma which has lagged far behind other disease. The data presented herein are robust and the approach is logical. The conclusions proposed are supported by the data presented. The analyses are robust and appear valid. The key references are included. No further work is requested as my prior comments have been adequately addressed.

Author's response

I thank this reviewer for these thoughtful comments. No further work has been advised

Reviewer #2 (Remarks to the Author):

The authors conducted multi-regional exome sequencing of 90 tumor biopsies from 22 malignant pleural mesothelioma patients, which includes 17 with an epithelioid and 5 with biphasic histology. They found that the most frequent mutations occurred in the BAP1 gene and they established that BAP1 and FBXW7 mutations were early driver mutations. NF2 mutations leading to Hippo pathway inactivation instead were late events. They found that very late NF2 mutations occurred in one patient. CDKN2A mutations were also relatively frequent. Instead, other gene mutations were not frequent. Overall the number of mutations was quite low, and copy number changes were relatively high. The study is quite important for the mesothelioma field although the cohort size is relatively small.

The authors addressed most of the questions in this revised manuscript. However, there is still an important issue that needs to be addressed in order to make an accurate conclusion. It is regarding the frequency of BAP1 mutation found in the somatic mesothelioma specimens. Although the authors stated that their reported frequency of 36% is similar to what was reported in the TCGA. And in the rebuttal letter, the authors also mentioned another two papers (Bueno et al Nature Genetics 2016, and Bott et al, Nature Genetics 2016- which in fact should be 2011) that had found similar results. Unfortunately, if the authors checked more carefully and thoroughly, they will find many more recent publications using comprehensive methods to check somatic alterations of BAP1 in mesothelioma tumor specimens, and the results from many research groups all confirm about 60% somatic alterations, for example, Nasu M et al J Thorac Oncol 2015. Hme Y et al. PNAS 2016; Lo lacono M et al, J Thorac Oncol 2015; Hmeljak J et al, Cancer Discovery 2018. Therefore, the authors should definitely think carefully about how to address this issue and make a more accurate statement.

Author's response

Agreed. We accept that independent studies have shown a higher rate of BAP1 somatic alterations in mesothelioma. To address this fact, a statement has been added in the discussion on page 12 (highlighted in yellow)

We reported a clonal BAP1 mutation rate of 36%, similar to that reported in the TCGA⁷. However, minute deletions revealed by single region high density comparative genomic hybridization array, combined with next generation sequencing implicates a generally higher rate of BAP1 somatic alterations in MPM³¹.

Reference 31 added. *Yoshikawa, Y. et al. High-density array-CGH with targeted NGS unmask multiple noncontiguous minute deletions on chromosome 3p21 in mesothelioma. Proc Natl Acad Sci U S A 113, 13432-13437, doi:10.1073/pnas.1612074113 (2016).*

Reviewer #3 (Remarks to the Author):

The authors have now satisfactorily addressed my comments. I have no further comments about this work. Good luck with publication.

Author's response

I thank this reviewer for these thoughtful comments. No further work has been advised